# A Four-Probiotics Regimen Combined with A Standard Helicobacter pylori-Eradication Treatment Reduces Side Effects and Increases Eradication Rates

**DOI:** 10.3390/nu14030632

**Published:** 2022-02-01

**Authors:** Nikos Viazis, Konstantinos Argyriou, Katerina Kotzampassi, Dimitrios K. Christodoulou, Periklis Apostolopoulos, Sotirios D. Georgopoulos, Christos Liatsos, Olga Giouleme, Kanellos Koustenis, Christos Veretanos, Dimitris Stogiannou, Miltiadis Moutzoukis, Charalambos Poutakidis, Ioannis Ioardanis Mylonas, Ioulia Tseti, Gerassimos J. Mantzaris

**Affiliations:** 1Gastroenterology Department, Evangelismos Hospital, 10676 Athens, Greece; k.koustenis@yahoo.gr (K.K.); christos.veretanos@yahoo.gr (C.V.); gjmantzaris@gmail.com (G.J.M.); 2Department of Gastroenterology, University Hospital of Larisa, 41334 Larissa, Greece; kosnar2@yahoo.gr; 3Endoscopy Unit, Department of Surgery, Aristotle University of Thessaloniki, 15341 Athens, Greece; kakothe@yahoo.com (K.K.); distorpd@windowslive.com (D.S.); 4Department of Gastroenterology, University Hospital of Ioannina, 45500 Ioannina, Greece; dchristodoulou@gmail.com (D.K.C.); miltiadismoutzoukis@gmail.com (M.M.); 5Gastroenterology Department, Army Share Fund Hospital (NIMTS), 11521 Athens, Greece; periclesapo@yahoo.com (P.A.); harrisp21@gmail.com (C.P.); 6Gastroenterology Department, Athens Medical, P. Faliro Hospital, 17562 Athens, Greece; georgpap@ath.forthnet.gr; 7Gastroenterology Department, 401 General Military Hospital of Athens, 11525 Athens, Greece; cliatsos@yahoo.com (C.L.); imylonas@yahoo.com (I.I.M.); 8Second Propedeutic Department of Internal Medicine, Medical School, Hippokration Hospital, Aristotle University of Thessaloniki, 54642 Thessaloniki, Greece; olga.giouleme@gmail.com; 9Uni-Pharma Kleon Tsetis Pharmaceutical Laboratories S.A., 14564 Athens, Greece; jtsetis@uni-pharma.gr

**Keywords:** Helicobacter pylori, non-bismuth quadruple eradication regimen, probiotics

## Abstract

Aim: To establish whether the addition of probiotics to a globally accepted Helicobacter pylori (H. pylori)-eradication scheme may reduce the rates of side effects and increase the eradication rates. Methods. Prospective, randomized, placebo-controlled trial of patients receiving eradication therapy for H. pylori in the eight participating centers. All patients received a 10-day proton pump inhibitor containing non-bismuth quadruple therapeutic regimen for H. pylori eradication (omeprazole 20 mg, amoxycillin 1 g, clarithromycin 500 mg, and metronidazole 500 mg all twice daily orally) and were randomized to receive either probiotics (group A) or placebo (group B). The probiotic used combined four probiotic strains, i.e., *Lactobacillus Acidophilus*, *Lactiplantibacillus plantarum*, *Bifidobacterium lactis,* and *Saccharomyces boulardii*. Results. Data were analyzed for 329 patients in group A and 335 patients in group B. Fifty six (17.0%) patients in group A and 170 (50.7%) patients in group B reported the occurrence of an H. pylori treatment-associated new symptom or the aggravation of a pre-existing symptom of any severity (*p* < 0.00001). H. pylori was successfully eradicated in 303 patients in group A (92.0%) and 291 patients in group B (86.8%), (*p* = 0.028). Conclusion: Adding probiotics to the 10-day concomitant non-bismuth quadruple H. pylori eradication regimen increases the eradication rate and decreases side effects.

## 1. Introduction

Helicobacter pylori (H. pylori) infection is considered the leading cause for the development of chronic active gastritis, gastroduodenalulcers, gastric mucosa-associated lymphoid tissue lymphoma, and gastric cancer [1]. As a result, National and International Guidelines have recommended several eradication regimens for H. pylori infection. Over the years, these regimens have been repeatedly modified to overcome the evolving resistance of H. pylori strains to antibiotics [2]. Recently a consensus meeting organized by the Hellenic Society of Gastroenterology recommended that a concomitant non-bismuth quadruple regimen, comprising of a proton pump inhibitor (PPI), amoxycillin, clarithromycin, and metronidazolefor at least 10 days, should be the ideal first-line therapy in Greece because this leads to higher H. pylori-eradication rates [3].

In 2001, an Expert Consensus of scientists from the Food and Agriculture Organization of the United Nations (FAO) and the World Health Organization (WHO) defined probiotics as “live microorganisms that, when administered in adequate amounts, confer a health benefit on the host” [4]. Based on the available literature, which includes well-designed clinical trials, systematic reviews, and meta-analyses, the consensus panel concluded that probiotics, as a general class, support a healthy digestive tract and a healthy immune system. This is mainly achieved by gut barrier reinforcement, neutralization of carcinogens, bile salt metabolism, vitamin synthesis, and enzymatic activity [4]. As regards H. pylori eradication, probiotics have been recommended in some current guidelines in an effort to improve adherence to and effectiveness of the eradication regimens [5]. Probiotics may competitively inhibit the colonization of H. pylori and produce bacteriostatic substances [6], while in addition, they enhance adherence to treatmentby reducing adverse events, such as diarrhea associated with antibiotics [7,8,9]. Nevertheless, other studies embrace opposite views on the efficiency of probiotics in assisting with the eradication [10,11].

Considering the contrasting views in the literature, we designed a prospective, multi-center, randomized, placebo-controlled trial to assess any potential effect of probiotics over placebo as additive agents to a globally standard H. pylori-eradication regimen. The primary objective of the study was to examine whether probiotics reduce the rates of adverse events associated with the recommended eradication regimen. The study’s secondary objective was to test whether adding probiotics increases the eradication rates.

## 2. Methods

### 2.1. Trial Design

This was a prospective, multi-center, placebo-controlled study that was performed in eight tertiary hospitals in Greece.

### 2.2. Patients

Consecutive patients, older than 18 years with a diagnosis of H. pylori infectionbetween January 2020 and September 2021 in each of the participating hospitals, were enrolled in this study. H. pylori infection was diagnosed according to the Maastricht V consensus criteria [12].

Exclusion criteria were (1) pregnancy or lactation, (2)prior therapy for H. pylori infection, (3) therapy with antibiotics or probiotics 1 month prior of entering the study, (4) therapy with proton pump inhibitors (PPIs), H2 receptor antagonists or antacids 2 weeks prior toentering the study, (5) known allergy to any of the antibiotics used in the eradication regimen, (6) prior diagnosis of heart disease, heart failure, malignancy, thyroid disease, lung disease, or any other disease that according to the treating physician precluded the patient from entering the study, and (7) denied written informed consent.

### 2.3. Diagnosis of H. pylori Infection

H. pylori infection was documented either with histological assessment (Hematoxyline—Eosin and Giemsa staining)ofgastric biopsies from the antrum and body during an upper gastrointestinal endoscopy or by a positive 13C-Urea Breath Test (UBT), (Helicobacter Test INFAI^®^, GmbH, Cologne, Germany).

### 2.4. Therapeutic Regimens

#### The study was conducted in a randomized, double-blind method

All patients received a 10-day PPI-containing quadruple H. pylori-eradication regimen consisting of oral omeprazole 20mg b.i.d., amoxycillin 1g b.i.d., clarithromycin 500mg b.i.d.,and metronidazole 500mg b.i.d.Omeprazole was taken orally 30–60 min prior, whereas all antibiotics were taken orally immediately after breakfast and dinner, respectively. In addition, patients were randomized to receive either a probiotic regimen (group A) or identical placebo (group B)twice daily, two hours after a meal for 15 days. Randomization was carried out via computer-generatedblock randomization list using a block size of four. Theprobiotic used (LactoLevure, Uni-Pharma S.A.—Athens—Greece) combined 4 probiotic strains, i.e., *Lactobacillus acidophilus*LA-5 (1,75Billion Colony Forming Units/capsule), *Lactiplantibacillus plantarum* (0.5Billion Colony Forming Units/capsule), *Bifidobacterium lactis*BB-12 (1.75Billion Colony Forming Units/capsule), and *Saccharomyces boulardii* (1.5 Billion Colony Forming Units/capsule). Stability studies on the finished product have been conducted in order to determine the effects of environmental conditions on product quality. Two different conditions (25 °C ± 2 °C/60% RH ± 5% and 40 °C ± 2 °C/75% RH ± 5%) were examined by LactoLevure to ensure product quality. The probiotic specimen and the identical placebo tablets were provided by Uni-Pharma S.A.—Greece.

### 2.5. Patients’ Follow Up

In all subjects included in the study, H. pylori eradication was tested with 13C-UBT 6 weeks after completion of the eradication regimen. Successful treatment was defined as a negative 13C-UBT. Compliance to treatment was assessed by counting the returned pills, and treatment adherence was defined as intake greater than 90%.

At the time of the follow up visit, all patients were asked to complete a questionnaire, to assess the occurrence of adverse events during the administration of the eradication regimen. Patients were asked to grade the occurrence of the following symptoms: epigastric pain, flatulence, early satiety, bitter taste, anorexia, nausea, vomiting, retrosternal burning, skin rash, and diarrhea on a scale of 0 to 3, where 0 was absence of symptoms, 1 was mild symptoms (transient and well tolerated), 2 was moderate symptoms (causing discomfort and partially interfering with common everyday activities), and 3 was severe symptoms (causing considerable interference with patients’ daily activities). Diarrhea was defined as at least three watery or loose stools per day for a minimum of 2 consecutive days [13]. Patients were asked to grade only the occurrence of new or aggravated symptoms during the administration of the eradication regimen relative to the baseline in order to minimize the regression-to-the-mean effect. They were also asked to grade the most prominent symptom in case they experienced more than one. Finally, all patients also fulfilled a questionnaire designed to assess the satisfaction with the treatment administered (GreekTreatment Satisfaction Questionnaire for Medication TSQM, version 1.4) [14].

### 2.6. Statistical Analysis

Chi-square, Fisher’s exact, and Student’s *t*-test were used to compare the demographic characteristics and frequencies of adverse events where appropriate. Statistical significance was defined as *p*-value less than 0.05.

In order to calculate the number of patients needed to be included in the study, we took into account the following two parameters: (i) that the Odds Ratio (OR) of successful H. pylori eradication with the co-administration of probiotics as opposed to placebo should be at least 2 and (ii) that we needed to have an appropriate statistical power of *p* = 0.90 also considering a type I error—α = 0.05). Given the above assumptions, we calculated that 288 patients should be allocated in each treatment group, giving a total of 576 patients to be randomized in a 1:1 ratio. Considering that in such studies, there is a large proportion (~20%) of subjects lost to follow-up, it was decided that the total number of patients to be initially enrolled should be 576/(100–20%) = 720 patients.

The primary and secondary outcomes of the study were the proportion of adverse events and the eradication rates. These outcomes were compared between the two treatment groups in 2 × 2 contingency tables reporting the *p*-value of the Fisher’s exact test and the effect size of the treatment through the OR with the 95% confidence intervals (CI). The analysis of the RCT was intended to be based on the intention-to-treat (ITT) principle. However, as it can be seen from Figure 1, 9.6% of the participants were lost to follow-up, and their outcomes were not recorded and remained unknown. Including these patients in the analysis would imply the unwarranted assumption that they all had the same outcome, therefore unjustifiably inflating the denominator, producing misleading inferences [15,16]. Consequently, it was decided that the analysis of the outcomes should be performed on the per-protocol (PP) basis, especially since the final number of the patients included was greater than the required sample size calculated with power analysis.

The study was approved by the ethics committee in all participating centers and was registered at clinical trials gov (NCT04178187).

The study was conducted according to good clinical practice guideline as well as the Declaration of Helsinki. All patients participating in the study signed a written consent form.

## 3. Results

The flow diagram of patient enrollment in the study isshown in Figure 1.

A total of 800 patients with H. pylori-positive chronic gastritis or peptic ulcer disease were screened in the participating centers during the study period. Fifty-nine patients did not meet the inclusion criteria and were excluded. Therefore, 741 patients were randomized to receive either 10-day omeprazole-containing quadruple therapy with probiotics (group A, *n* = 371) or 10-day omeprazole-containing quadruple therapy with placebo (group B, *n* = 370). The baseline characteristics and the endoscopic findings of the study population can be seen in Table 1 and Table 2, respectively. Four patients (all in group B) did not complete the eradication regimen due to adverse events, i.e., diarrhea (*n* = 3) and vomiting (*n* = 1). Furthermore, 42 patients in group A and 31 patients in group B, respectively, were lost to follow up. These patients completed the eradication regimen to which they were allocated, but they were not checked for successful eradication. Therefore, data were analyzed for 329 patients in group A and 335 patients in group B. After counting the pills, all patients included in the study were considered adherent to treatment.

### 3.1. Adverse Events

During the administration of the eradication regimen, 56 (17.0%) patients in group A and 170 (50.7%) patients in group B reported the occurrence of an H. pylori treatment-associated new symptom or the aggravation of a pre-existing symptom of any severity (*p* < 0.00001). All symptoms reported by the patients can be seen in Table 3.

Odds ratio (95% confidence intervals) were 0.072 (0.026–0.202). The reverse of the OR of 0.072 equals to 13.9. This means that the ratio of adverse events to no adverse events is almost fourteen times less in the probiotic compared to the placebo arm.

As regards the severity of symptoms, 4 patients in group A (1.2%) and 49 patients in group B (14.6%) reported symptoms with a severity of grade 3 (*p* < 0.00001), whereas 52 patients in group A (15.8%) and 121 patients in group B (38.2%) reported symptoms with a severity of grade 1 or 2 (*p* < 0.00001).

Treatment satisfaction (mean (SD))as reported by the TSQM questionnaire was 85.6 (11.9) in Group A and 75.2 (12.6) in group B (*p* < 0.001).

### 3.2. Eradication Rates

H. pylori was successfully eradicated in 303 patients in group A (92.0%) and 291 patients in group B (86.8%), (*p* = 0.028). In the probiotics group, the odds of eradication to no-eradication were 303/26 = 11.65. In the placebo group, these odds were 291/44 = 6.61. This yields an odds ratio: OR = 11.65/6.61 = 1.76 (95% CI 1.06–2.94). This effect size shows that the use of probiotics almost doubles the ratio of eradication to no-eradication in comparison to the use of placebo.

## 4. Discussion

Τhis prospective, randomized, placebo-controlled study aimed at investigating the effects of probiotic therapy on the development of symptoms associated with an internationallyrecommendedH. pylori-eradicationregimen. Our results show that a twice daily intake of probiotic supplementation (LactoLevure, Uni-pharma S.A.—Greece) significantly reduced the onset of new symptomsassociated with the H. pylori-eradication regimen or the aggravation of pre-existing symptoms of any severity. Adding probiotics to the standard H. pylori-eradication regimen also increased the eradication rates (92.0% in group A vs. 86.8%in group B, PP analysis, *p* = 0.028).

Since its discovery in 1983, H. pylori has been extensively investigated and has gained interest in gastrointestinal oncology due to its well-established causative role in the pathway of gastric carcinogenesis [17]. Its eradication can resolve gastritis and may substantially reduce the incidence of gastroduodenal ulcers, low-grade MALT lymphomas, and gastric adenocarcinomas. According to recent National guidelines, a concomitant non-bismuth quadruple regimen for at least 10 days is currently the ideal treatment therapy for the eradication of H. pylori in Greece since it provides high eradication rates [2,3]. Non-bismuth quadruple regimens have shown higher efficacy in many trials and are at present a feasible option, especially in countries like ours, where bismuth is not available [18,19,20]. This finding was also shown in two prospective randomized trials that compared concomitant and sequential regimens of the same duration (10 days). In both trials, the concomitant regimens showed significantly better results as compared to the sequential ones [21,22]. The eradication rates observed in our study, in which we used such a concomitant, non-bismuth quadruple regimen for 10 days, were 92% and 86.8% in groups A and B, respectively, in accordance with the rates reported so far in our country.

Despite the high eradication rates reported in clinical trials, two major drawbacks, namely antibiotic resistance and poor compliance, may hinder the satisfactory effects of standard eradication schemes [23,24]. Poor compliance has been associated with the side effects during treatment, such as antibiotic-associated diarrhea, nausea, or vomiting. According to literature data, side effects associated with eradication therapy may appear in 5 to 30% of cases and may lead to treatment discontinuation [25,26]. Probiotics can potentially alleviate these side effects and boost H. pylori-eradication rates [27,28,29,30]. These characteristics make probiotics a promising adjuvant treatment to the standard H. pylori-eradication regimens. Probiotics have several natural advantages, such as safety and an anti-pathogen ability, whereas certain probiotic strains, such as *Lactobacillus* spp., can colonize the gastric microbial environment, directly or indirectly antagonizing H. pylori [31,32,33]. Furthermore, short-chain fatty acids produced by probiotics may reduce urease activity and lead to inhibition of the H. pyloricolonization in the gastric mucosa [34]. Finally, probiotics have a positive effect on inhibiting the inflammatory response mediated by interleukin (IL)-8 after an H. pylori infection [35]. The probiotic used in our study (LactoLevure, Uni-pharma S.A.—Greece) combines four probiotic strains, i.e., (1) *Lactobacillus Plantarum* (UBLP 40)new name:*Lactiplantibacillusplantarum*, (2) Lactobacillus acidophilus (LA-5^®^), (3) *Bifidobacterium animalis*subsp. *Lactis* (BB-12^®^), and (4) Saccharomycesboulardii Unique-28, and has been successfully administered for the prevention of ventilator associated pneumonia in multi-trauma patients [36] for the treatment of irritable bowel syndrome [37] and for the prevention of post-operative complications after colorectal surgery [38]. These data, together with the fact that LactoLevure contains a sufficient amount of *Lactobacillus* spp., gave us the rationale to use this specific probiotic specimen in our study. Far from these, the optimal duration for probiotic supplementation has not been adequately defined; however, it seems that a period of at least 2 weeks is needed for maximizing positive effects [39]. Based on the aforementioned data, we decided to administer the probiotic supplementation for 15 days.

Despite the favorable properties attributed to probiotics, according to the 2016 Toronto consensus, there are no sufficient data proving that the addition of probiotics does increase the H. pylori eradication rates and reducestreatment-associated side effects [40]. A network meta-analysis, which included 40 eligible studies with 894 patients, concluded that compared to the control group a higher eradication rate (*p* < 0.001) and lower incidence of total side effects (*p* < 0.001) were observed in the probiotic group [39]. Similarly, the 2017 ACG clinical guideline, based on data from a meta-analysis, stated that probiotics do increase H. pylori-eradication rates and reduce the occurrence of side effects. However, criticisms have been raised due to the fact that thetrials included in the meta-analysis werecarried out in China, and a high risk of bias could not be excluded [5]. Thus, currently, there is no conclusion about the best choice of probiotics as well as the dose and course of treatment. Our data provide further evidence that probiotics can indeed increase H. pylori-eradication rates and decrease side effects. Our study was adequately powered and included a homogenous Caucasian population, which was treated for all the spectrum of gastric diseases that need H. pylori eradication. Although our study was designed prior to the appearance of COVID-19, it was executed during the pandemic. We therefore believe that given the unfavorable circumstances, the number of patients lost to follow up was relatively small. Furthermore, and according to our knowledge, our study is the first one to examine in a prospective randomized way the effect of probiotics in adjunct to the concomitant quadruple non-bismuth H. pylori-eradication regimen, which is the standard practice today in many countries, including Greece. This is of particular importance since in the abovementioned meta-analysis, probiotics were shown to improve the eradication rates and decrease side effects when administered with bismuth-containing quadruple eradication regimens [39]. Our study was also the first to grade the severity of side effects and analyze the effect of probiotics in both severe and mild-to-moderate adverse reactions. Taste disturbances and diarrhea were by far the most common side effects. Both severe and non-severe adverse reactions were statistically less common in the group of patients receiving probiotics, and this is most probably the reason of greater satisfaction to treatment in this specific group of patients. On the other hand, the main limitation of our study is that we did not evaluate antibiotic resistance in the studied patients; however, arecent publication in our country gives a good estimation of the expected resistance inthe treated population [21]. Moreover, it is logical to hypothesize that resistant strains have been allocated equally to the two treatment arms due to the randomized design of our study. Finally, we did not re-endoscope our patients after the probiotic consumption since we used a 13C-UBT to test H. pylori eradication, and that is why we only present endoscopic data only at patients’ entry into the study.

In conclusion, our prospective, randomized, placebo-controlled trial showed that adding probiotics to the 10-day concomitant non-bismuth quadruple H. pylori-eradication regimen increases the eradication rate and decreases side effects.

## Figures and Tables

**Figure 1 nutrients-14-00632-f001:**
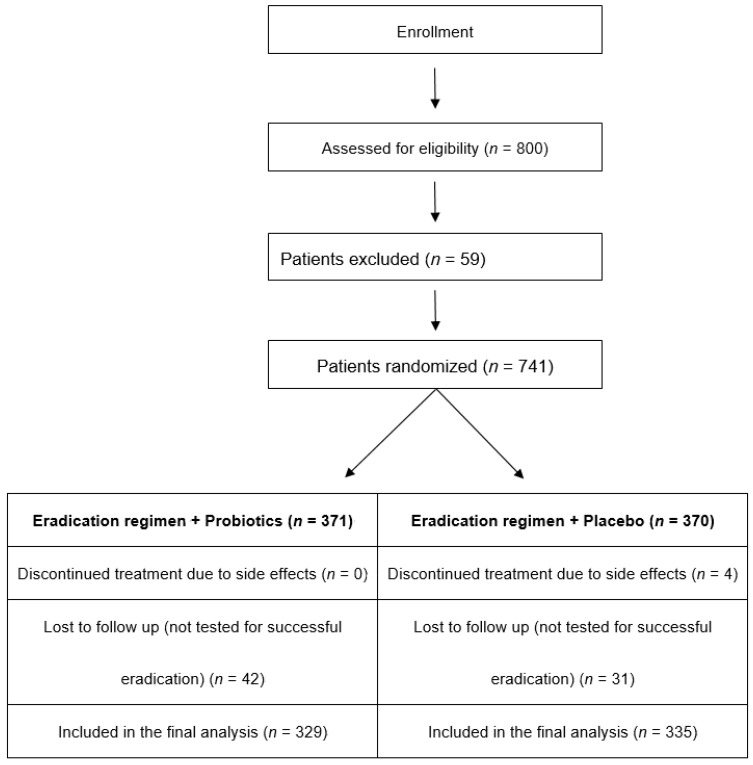
Flow diagram of patients included in the study.

**Table 1 nutrients-14-00632-t001:** Baseline characteristic of the study population.

Variable	Group A	Group B	*p*-Value
Age, years (SD)	51.6 (15.9)	49.9 (16.3)	NS
Sex, male/female	183/188	178/192	NS
Smoking habits	129 (34.7%)	141 (38.1%)	NS
Alcohol intake	22 (5.9%)	19 (5.1%)	NS
Family history of gastric cancer	2 (0.5%)	1 (0.3%)	NS
BMI (Kg/m^2^)	28.6 (5.3)	27.9 (6.9)	NS

BMI = Body Mass Index, NS = Non Significant.

**Table 2 nutrients-14-00632-t002:** Endoscopic findings of the study population.

Variable	Group A	Group B	*p*-Value
Peptic ulcer	*n* = 45(12.1%)	*n* = 37 (10.0%)	NS
Gastritis without gastric atrophy	*n* = 148 (39.9%)	*n* = 154 (41.6%)	NS
Gastritis with gastric atrophy	*n* = 122 (32.9%)	*n* = 130 (35.1%)	NS
Gastritis with intestinal metaplasia	*n* = 56 (15.1%)	*n* = 49 (13.2%)	NS

**Table 3 nutrients-14-00632-t003:** Occurrence ofsymptoms as reported by patients in both groups during the administration of the eradication regimen.

Variable	Group A	Group B	*p*-Value
Epigastric pain	10	22	0.03
Flatulence	2	19	0.0001
Early satiety	1	9	0.01
Bitter taste	20	36	0.03
Anorexia	0	1	-
Nausea	3	22	0.0001
Vomiting	1	7	0.03
Retrosternal burning	7	19	0.01
Skin rash	0	0	-
Diarrhea	12	35	0.0006

## Data Availability

Not applicable.

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
