# Peer review of "A Four-Probiotics Regimen Combined with A Standard Helicobacter pylori-Eradication Treatment Reduces Side Effects and Increases Eradication Rates"

_nutrients, 2022, doi:10.3390/nu14030632_

Round 1
Reviewer 1 Report
The authors presented the results of clinical trial of the combination of probiotics in Helicobacter pylori (H. pylori) eradication. Some revisions should be done before reconsidering for publication.
- The scientific name of Lactobacillus Plantarum should be revised as Lactiplantibacillus plantarum. All the bacteria names should be presented in italic.
- The authors should present the Endoscopic findings of the study population before and after probiotic consumption.
- The descriptions of "This effect size shows that the use of probiotics almost doubles the ratio of eradication to no-eradication in comparison to the use of placebo." were questionable. From the data provided, the successful H. pylori eradication in the probiotics consumption group or placebo group was 92% or 86.8%, not double.
- For the probiotics formula, the full name of "BU" should be clearly provided. (line 93 to 95)
- The rationale for choosing this LactoLevure with 4 probiotics combination to examine the beneficial effect in H. pylori eradication should be clearly provided.
Author Response
Comment: The scientific name of Lactobacillus Plantarum should be revisedas Lactiplantibacillus plantarum. All the bacteria names should be presented in italic.
Reply: We have revised the name of the bacteria as suggested and have presented all bacteria names in italics.
Comment: The authors should present the Endoscopic findings of the study population before and after probiotic consumption.
Reply: The endoscopic findings of the study population before the probiotic consumption (at entry into the study) is presented in table 2. All patients that concluded the study were tested for h. pylori eradication with a 13C-urea breath test and were not re-endoscoped. That is why we did not report the endoscopic findings of the study population after the probiotic consumption. This information has been added at the Discussion section as one of the limitations of the study.
Comment: The descriptions of "This effect size shows that the use of probiotics almost doubles the ratio of eradication to no-eradication in comparison to the use of placebo." were questionable. From the data provided, the successful H. pylori eradication in the probiotics consumption group or placebo group was 92% or 86.8%, not double.
Reply: As mentioned in the statistical analysis section the effect size of the treatment was expressed through the OR. This was expressed in the following manner: In the treatment group the odds of eradication to no-eradication were 303/26 = 11.65. In the placebo group these odds were 291/44 = 6.61. Therefore, the odds ratio was OR = 11.65/6.61 = 1.76, which is nearer to the integer 2.
This information has been added in the Results section.
Comment: For the probiotics formula, the full name of "BU" should be clearly provided. (line 93 to 95)
Reply: BU has been changed to Colony Forming Units according to the suggestion of Reviewer 2.
Comment: The rationale for choosing this LactoLevure with 4 probiotics combination to examine the beneficial effect in H. pylori eradication should be clearly provided.
Reply: The probiotic used in our study (Lactolevure, Uni-pharma S.A. – Greece) combines 4 probiotic strains, i.e., Saccharomyces Boulardii, Bifidobacterium Lactis, Lactobacillus Acidophilus &Lactiplantibacillus plantarum and has been successfully administered for the prevention of ventilator associated pneumonia in multi-trauma patients, for the treatment of irritable bowel syndrome and for the prevention of post-operative complications after colorectal surgery. These data, together with the fact that Lactolevure contains a sufficient amount of Lactobacillus spp. (which have been proven beneficial for H. pylori eradication) gave us the rationale to use this specific probiotic specimen in our study.
This information has been added at the Discussion section.
Reviewer 2 Report
The manuscript is well written although I have a series of doubts that I need to clarify.
There are minor typos or curiosities, that should be revised, explained and, if necessary, corrected.
Minor points:
The introduction is very short. It does not describe what the side effects of quadruple therapy are. The introduction discusses the concept of probiotics. It does not explain what probiotics are or their functionality. I recommend the latest consensus on probiotics:
- FAO; WHO. Probiotics in Food: Health and Nutritional Properties and Guidelines for Evaluation; FAO Food and Nutrition Paper; Food and Agriculture Organization of the United Nations: Rome, Italy; World Health Organization: Geneva, Switzerland, 2006; 50p.
- Hill, C.; Guarner, F.; Reid, G.; Gibson, G.R.; Merenstein, D.J.; Pot, B.; Morelli, L.; Canani, R.B.; Flint, H.J.; Salminen, S.; et al. Expert Consensus Document: The International Scientific Association for Probiotics and Prebiotics consensus statement on the scope and appropriate use of the term probiotic. Nat. Rev. Gastroenterol. Hepatol. 2014, 11, 506–514.
Please could you expand slightly on the introduction?
On line 50, I do not understand (are QPS?) "fecal bacteria" and “Bacillus licheniformis”. The references [4, 5] cite them?. The authors should revise the writing of the different probiotic species that appear in the text (in italics and the letter of the species in lower case).
In lines 93 to 95, I suppose that the concentration of microorganisms is expressed, but I do not understand the meaning of BU. Wouldn't CFU (colony forming units) be more appropriate?
In the discussion, probiotic strains are named but, only species are written. can the authors cite the specific strains involved in the study?
Is this study funded or sponsored by a pharmaceutical laboratory? There is a co-author who works at Uni-Pharma and the product used is from that laboratory. Could there not be a conflict of interest?
Was the stability of the probiotic mixture tested?
What has been the participation of each author in the study?
The authors believe that, by using these 4 probiotic species even if it was not the LactoLevure product, the same results would be obtained? If the answer is yes, then would it be the same to use this product or another one with the same species?
Author Response
Comment: The introduction is very short. It does not describe what the side effects of quadruple therapy are. The introduction discusses the concept of probiotics. It does not explain what probiotics are or their functionality. I recommend the latest consensus on probiotics:
- FAO; WHO. Probiotics in Food: Health and Nutritional Properties and Guidelines for Evaluation; FAO Food and Nutrition Paper; Food and Agriculture Organization of the United Nations: Rome, Italy; World Health Organization: Geneva, Switzerland, 2006; 50p.
- Hill, C.; Guarner, F.; Reid, G.; Gibson, G.R.; Merenstein, D.J.; Pot, B.; Morelli, L.; Canani, R.B.; Flint, H.J.; Salminen, S.; et al. Expert Consensus Document: The International Scientific Association for Probiotics and Prebiotics consensus statement on the scope and appropriate use of the term probiotic. Nat. Rev. Gastroenterol. Hepatol. 2014, 11, 506–514.
Please could you expand slightly on the introduction?
Reply: We have expanded slightly the introduction, according to your suggestion, using the Consensus document by Hill et al.
Comment: On line 50, I do not understand (are QPS?) "fecal bacteria" and “Bacillus licheniformis”. The references [4, 5] cite them?.
Reply: We have deleted these words and re-written the phrase to avoid confusion. We have also deleted reference 4 and replaced it with the consensus for the probiotics.
Comment: The authors should revise the writing of the different probiotic species that appear in the text (in italics and the letter of the species in lower case).
Reply: We have re-written the names of the probiotic species as suggested
Comment: In lines 93 to 95, I suppose that the concentration of microorganisms is expressed, but I do not understand the meaning of BU. Wouldn't CFU (colony forming units) be more appropriate?
Reply: BU means Billion Units. BU and CFU are the same thing. We have substituted BU with CFU at the text according to your suggestion
Comment: In the discussion, probiotic strains are named but, only species are written. can the authors cite the specific strains involved in the study?
Reply: The specific strains involved in the study are as following.
1)Lactobacillus Plantarum (UBLP 40) à new name : Lactiplantibacillus plantarum
2)Lactobacillus acidophilus (LA-5®)
3)Bifidobacterium animalis subsp. lactis (BB-12®)
4)Saccharomyces boulardii Unique-28
These data have been added at the Discussion.
Comment: Is this study funded or sponsored by a pharmaceutical laboratory? There is a co-author who works at Uni-Pharma and the product used is from that laboratory. Could there not be a conflict of interest?
Reply: The probiotic regimen used in the study was LactoLevure which is indeed produced by Uni-Pharma. The pharmaceutical company provided us the LactoLevure tablets, as well as the identical placebo tablets used (this information has been added at the Methods). That was their only contribution in the study and has been declared in the authors’ contribution as funding acquisition (since there was no other way to describe it). The pharmaceutical company was not involved in any other way in the design or execution of the study, and we therefore believe that there is no conflict of interest.
Comment: Was the stability of the probiotic mixture tested?
Reply: Stability studies on the finished product have been conducted in order to determine the effects of environmental conditions on product quality. Two different conditions (25 °C ± 2°C / 60% RH ± 5% and 40 °C ± 2°C / 75% RH ± 5%) were examined on specified time intervals and all the data are available in the technical file of the product. This information has been added at the Methods section.
Comment: What has been the participation of each author in the study?
Reply: The participation of each author in the study can be seen below. This information has been provided to Nutrients and has been uploaded together with the manuscript at the submission site.
Conceptualization, Nikos Viazis and Gerassimos J Mantzaris; Data curation, Nikos Viazis, Kostas Argyrios, Katerina Kotzampassi, Dimitrios Christodoulou, Periklis Apostolopoulos, Sotirios D Georgopoulos, Christos Liatsos, Olga Giouleme, Kanellos Koustenis, Christos Veretanos, Dimitris Stogiannou, MiltiadisMoutzoukis, CharalambosPoutakidis and Ioannis Mylonas; Formal analysis, Nikos Viazis, Sotirios D Georgopoulos and Christos Liatsos; Funding acquisition, IouliaTseti; Investigation, Nikos Viazis; Methodology, Nikos Viazis, Kostas Argyrios, Katerina Kotzampassi, Dimitrios Christodoulou and Periklis Apostolopoulos; Writing – original draft, Nikos Viazis; Writing – review & editing, Kostas Argyrios, Katerina Kotzampassi, Dimitrios Christodoulou, Periklis Apostolopoulos, Sotirios D Georgopoulos, Christos Liatsos, Olga Giouleme and Gerassimos J Mantzaris.
Comment: The authors believe that, by using these 4 probiotic species even if it was not the LactoLevure product, the same results would be obtained? If the answer is yes, then would it be the same to use this product or another one with the same species?
Reply: We believe that same results would be obtained if another product with the same species was used; however, there is no other product with the same species in the Greek market.
Reviewer 3 Report
The manuscript entitled "A FOUR-PROBIOTICS REGIMEN COMBINED WITH A 2 STANDARD HELICOBACTER PYLORI ERADICATION 3 TREATMENT REDUCES SIDE EFFECTS AND INCREASES 4 RATES OF ERADICATION" deals with the subject concerned, unfortunately H. pylori infections are very widespread, often also experienced if not treated promptly and adequately, in fact it can lead to permanent infections, generate gastric ulcers, gastroesophageal reflux.
The manuscript is well structured, the results are interesting.
However, I have some suggestions for the authors to dare:
In the discussion it would be necessary to ask as previous studies have question as LPS essential component of H. pylori goes to act of inflammation COX-2 marker of inflammation (doi: 10.13. Assays have been made to evaluate the markers and / or biomarkers inflammation? If you and would be introduced into the results.
In addition, if in the interesting discussion the data obtained with in the studies introduced the concept of the importance of antimicrobial pept in the treatment of H. pylori and in the micromiota of affected patients (doi: 10.393 / biom9060237; doi: 10.171 / journal .pone. 0222295).
Author Response
Comment: In the discussion it would be necessary to ask as previous studies have question as LPS essential component of H. pylori goes to act of inflammation COX-2 marker of inflammation (doi: 10.13. Assays have been made to evaluate the markers and / or biomarkers inflammation? If you and would be introduced into the results.
Reply: Thank you for the comment. We did not test any markers of inflammation. This was out of the scope of this study which was aiming to examine the role of probiotics as an adjunct therapy to the standard H. pylori eradication regimens.
Comment: In addition, if in the interesting discussion the data obtained with in the studies introduced the concept of the importance of antimicrobial pept in the treatment of H. pylori and in the micromiota of affected patients (doi: 10.393 / biom9060237; doi: 10.171 / journal .pone. 0222295).
Reply: Unfortunately, we did not study antimicrobial peptides or the microbiota of the affected patients, since – once again – this was out of the scope of our study.
Round 2
Reviewer 1 Report
The authors have answered my questions and the manuscript can be accepted in the present form.
Author Response
Thank you